# Comparison of antibody responses to SARS-CoV-2 variants in Australian children

Zheng Quan Toh [1,2,5], Nadia Mazarakis [1,5], Jill Nguyen[1], Rachel A. Higgins[1], Jeremy Anderson[1,2], Lien Anh Ha Do [1,2], David P. Burgner [1,2,3], Nigel Curtis[1,2,3], Andrew C. Steer[1,2,3], Kim Mulholland[1,2,4], Nigel W. Crawford[1,2,3], Shidan Tosif[1,2,3,6] & Paul V. Licciardi [1,2,6] ✉

There is limited understanding of antibody responses in children across different SARS-CoV-2 variants. As part of an ongoing household cohort study, we assessed the antibody response among unvaccinated children infected with Wuhan, Delta, or Omicron variants, as well as vaccinated children with breakthrough Omicron infection, using a SARS-CoV-2 S1-specific IgG assay and surrogate virus neutralization test (% inhibition). Most children infected with Delta (100%, 35/35) or Omicron (81.3%, 13/16) variants seroconverted by one month following infection. In contrast, 37.5% (21/56) children infected with Wuhan seroconverted, as previously reported. However, Omicron-infected children (geometric mean concentration 46.4 binding antibody units/ml; % inhibition = 16.3%) mounted a significantly lower antibody response than Delta (435.5 binding antibody untis/mL, % inhibition = 76.9%) or Wuhan (359.0 binding antibody units/mL, % inhibition = 74.0%). Vaccinated children with breakthrough Omicron infection mounted the highest antibody response (2856 binding antibody units/mL, % inhibition = 96.5%). Our findings suggest that despite a high seropositivity rate, Omicron infection in children results in lower antibody levels and function compared with Wuhan or Delta infection or with vaccinated children with breakthrough Omicron infection. Our data have important implications for public health measures and vaccination strategies to protect children.

Children have been less likely to be infected and develop severe disease by the original SARS-CoV-2 (Wuhan) strain compared to adults[1–3]. However, the combination of increased transmissibility of SARS-CoV-2 Delta and Omicron variants, increased population movement due to the easing of COVID-19 restrictions, and a higher vaccination rate in adults compared with children have resulted in rising COVID-19 cases among children[4,5]. Despite this, SARS-CoV-2 infections in children are mostly mild or asymptomatic.

The Omicron variant BA.1/BA.2 is associated with reduced clinical severity and risk of hospitalization compared to the Delta variant in both children and adults[6–9]. Adults mount strong Omicron-specific humoral responses[10], but limited data are available in children. We previously reported that only 37.5% of children infected with the Wuhan strain seroconverted compared with 76.2% of adults[11]. It is unknown if a similar pattern occurs following Delta or Omicron infection in children.

[1]Infection and Immunity, Murdoch Children's Research Institute, Parkville, VIC, Australia. [2]Department of Paediatrics, The University of Melbourne, Parkville, VIC, Australia. [3]Department of General Medicine, The Royal Children's Hospital, Parkville, VIC, Australia. [4]Faculty of Epidemiology and Public Health, London School of Hygiene and Tropical Medicine, London, UK. [5]These authors contributed equally: Zheng Quan Toh, Nadia Mazarakis. [6]These authors jointly supervised this work: Shidan Tosif, Paul V. Licciardi. ✉e-mail: paul.licciardi@mcri.edu.au

In Australia, there have been three epidemic waves of COVID-19, caused respectively by the original Wuhan strain (first infection documented in March 2020), the Delta variant (May 2021), and the Omicron variant (November 2021)[12]. Australian children aged 5–11 years were eligible for COVID-19 vaccination from December 2021, with uptake of two doses estimated at 40.6% as of 7th August 2022[5]. Here, we show seropositivity (seroconversion) rates and antibody responses in children across the three SARS-CoV-2 waves in Melbourne, Australia.

## Results

Between March 2020 and July 2022, a total of 580 adults and children were enroled. Participants aged between 6 months to 17 years with COVID-19 confirmed by SARS-CoV-2 PCR or rapid antigen test (RAT) on nasopharyngeal swab who had not received any COVID-19 vaccine were included in this analysis ($n = 107$). We also included children who had received at least one dose of a COVID-19 vaccine but experienced a breakthrough Omicron infection ($n = 24$, Supplementary Fig. 1 and Supplementary Table 1). Unvaccinated children infected with Omicron were younger than children infected with Wuhan or Delta variant

Among the 56 children infected with the Wuhan strain, 21 (37.5%) seroconverted by day 36[11], while all 35 children infected with Delta ($P < 0.0001$), and 13 of 16 (81%) infected with Omicron ($P = 0.004$) variants seroconverted based on testing using the Wuhan S1 antigen (Fig. 1A). Two out of three children infected with Omicron were seronegative based on testing using the Wuhan S1 antigen but were seropositive when tested using the Omicron S1 antigen.

Among children who seroconverted, S1-specific IgG concentrations to Omicron infection (GMC: 46.4 BAU/ml, 95% CI: 18.9, 113.8) were 7.7- and 9.4-fold lower than Wuhan (GMC: 359 BAU/ml, 95% CI: 221.2, 582.5, $P = 0.0002$) or Delta infections (GMC: 435.5 BAU/ml, 95% CI: 296.9, 638.9, $P < 0.0001$), respectively (Fig. 1B). Vaccinated children with Omicron breakthrough infection had 61.6-fold higher IgG GMC than unvaccinated Omicron-infected children. Similarly, lower neutralizing antibody responses based on the Wuhan-antigen-based sVNT were observed among unvaccinated Omicron-infected children (only one out of the 13 children who seroconverted had a positive neutralizing antibody response) compared with children who seroconverted following Wuhan or Delta infection or who were previously vaccinated (19 of 21, 35 of 35 and 24 of 24 had high neutralizing antibody responses respectively, $P < 0.0001$ for all

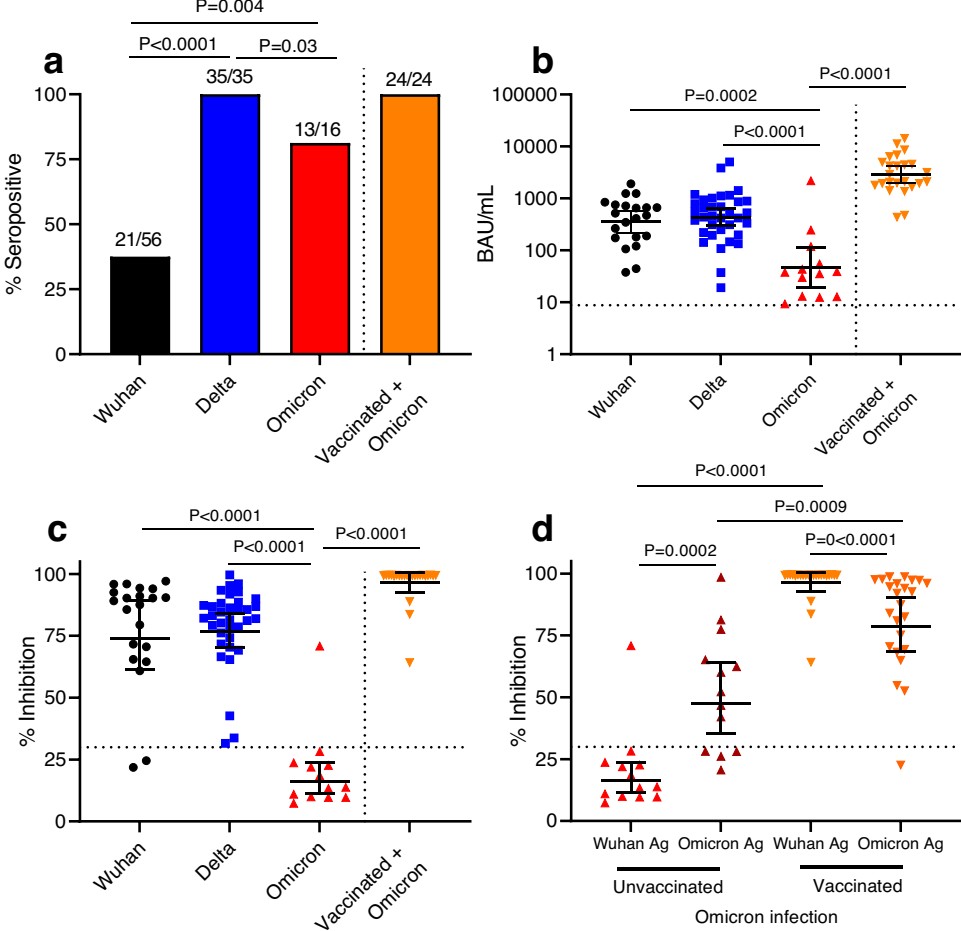

**Fig. 1 | SARS-CoV-2 antibody responses at Day 36 post-infection in children infected with Wuhan (black), Delta (blue), or Omicron (red) variant and children previously vaccinated and then infected with Omicron (orange). a** SARS-CoV-2 Immunoglobulin G (IgG) Seropositivity Rate measured by IgG ELISA using the Wuhan S1 antigen and compared using Fisher's Exact test. **b** SARS-CoV-2 IgG concentrations as measured by ELISA using the Wuhan S1 antigen. **c** SARS-CoV-2 neutralizing antibodies as measured by surrogate virus neutralization test using the Wuhan-specific RBD antigen. Comparison of S1-specific IgG antibody concentrations and neutralizing antibodies (% inhibition) between children from different

SARS-CoV-2 waves, between unvaccinated and vaccinated were done using a two-sided Mann–Whitney U test. **d** SARS-CoV-2 neutralizing antibodies as measured by surrogate virus neutralization test using Wuhan RBD- or Omicron RBD-specific antigen; variant-specific antibody responses were compared using a two sided Wilcoxon-signed rank test. Dotted lines indicate seropositivity cut-off. Each data point represents a seropositive individual participant for each wave (Wuhan, $n = 21$; Delta, $n = 35$; Omicron, $n = 13$) and for those vaccinated and then infected ($n = 24$). Data presented as geometric mean concentrations ± 95% confidence intervals. BAU, binding antibody units. Source data are provided as a Source Data file.

comparisons, Fig. 1C). This lower response to Omicron was not associated with younger age when we stratified the analysis based on 5–17 years and <5 years for Wuhan and Delta (Supplementary Fig. 2). When we used variant-specific S1 antigens, we observed higher IgG concentrations for Omicron but not Delta (Supplementary Fig. 3). Omicron-specific neutralizing antibody responses were detected in nine out of 13 children when measured using the Omicron-specific sVNT for the Omicron cohort (geometric mean % inhibition: 47.7%, 95% CI 35.4%, 64.2%), but the responses were still significantly lower compared with vaccinated children with Omicron breakthrough 78.8% (95% CI 68.7%, 90.4%, $P = 0.009$, Fig. 1D). Children infected with Wuhan or Omicron, but not Delta, had significantly lower IgG concentrations than adults (2.1-fold and 89.1-fold, respectively), although the number of unvaccinated adults in the Omicron cohort was small due to high vaccine coverage among adults in the cohort (Supplementary Fig. 4). The neutralizing antibody response against Omicron was consistent with the IgG response (Supplementary Fig. 4).

## Discussion

Our findings indicate that almost all children seroconverted following Omicron infection. However, these children mounted a lower antibody response compared to the Wuhan strain or the Delta variant, which contrasts with observations in adults[10]. How this relates to protection against re-infection and disease is unknown. The role of other immune factors (i.e., cellular and mucosal immunity) in protection against Omicron re-infection, particularly BA.4/5 subvariants among unvaccinated and vaccinated children remains to be determined.

The lower seroconversion rate among children infected with Wuhan strain was previously reported[11]. It was suggested that a majority of children were protected through a combination of enhanced innate immune and local (mucosal) immune responses which cleared the virus more efficiently, preventing the establishment of infection and subsequent seroconversion. A recent paper also documented lower antibody responses in children infected with Wuhan strain compared with adults with mild or asymptomatic COVID-19 in line with these findings[13]. The high seroconversion rate in children infected with Delta or Omicron is likely due to the higher viral loads previously reported for these variants[14,15]. Consequently, the differences in antibody response may also be due to the viral load and the duration of shedding or differences in clinical severity. However, we were unable to quantify the viral load in children infected with Delta or Omicron variants for this study while children in our cohort were non-hospitalized and had mild/asymptomatic infection.

Vaccinated children with breakthrough Omicron infection mounted the highest antibody responses in our study, suggesting that vaccination may help overcome the low antibody response following Omicron variant infection in children. We also found that the antibody response against Omicron may be substantially underestimated when measured using Wuhan antigen, with implications for seroprevalence studies and estimating protective antibody titres against Omicron infection. On 3rd August 2022, Australia authorities recommended the use of the mRNA-1273 (Moderna®) vaccine for high-risk children aged from 6 months to <5 years[16]. Ongoing serological studies to monitor immunogenicity against SARS-CoV-2 variants of concern remains crucial. Furthermore, new formulations of vaccine(s) are under review by international regulators, including bivalent Ancestral/Omicron vaccines[17], and our household cohort study will continue to monitor immunological responses when these new-generation vaccines become available to the paediatric population. Limitations of the study include the small sample size, particularly for the Omicron cohort and the young age group (<5 years) in the Omicron cohort, since most children in our cohort were ≥5 years and had received at least one dose of COVID-19 vaccine. We did not

have pre-infection blood samples to rule our prior infection but we believe this to be unlikely given the low SARS-CoV-2 exposure reported among Australian children prior to Omicron infection[18]. We also did not have data on viral loads and information on the infecting strain. However, we believe our assumptions about the infecting variant in each cohort were correct based on available local epidemiological and sequence data[12].

In summary, the immune response to SARS-CoV-2 infection among children varies between the variants. While most unvaccinated children infected with Delta or Omicron seroconvert, antibody responses to Omicron were much lower. This might lead to an increased risk of repeated re-infection of children, which would in-turn have serious implications for their long-term health. The much stronger antibody responses observed among vaccinated children supports vaccination of children to induce greater protective immunity.

## Methods

The study was conducted with the approval of the Royal Children's Hospital Human Research Ethics Committee (HREC): HREC/63666/RCHM-2019. Written informed consent and assent were obtained from adults/parents and children, respectively.

### Epidemic waves

As of July 2022, Australia has had three epidemic waves of COVID-19 caused by the original (ancestral) Wuhan strain (first infection documented in March 2020), Delta (May 2021) and Omicron (November 2021) variants. In Victoria, Australia, the Delta and Omicron variant was first detected in late May 2021 and early December 2021, respectively. The Delta cohort was recruited between July 2021 and November 2021, while the Omicron cohort was recruited between January 2022-July 2022 based on data from Nextstrain.org.

### Study design

In this case-ascertained study, suspected SARS-CoV-2 cases and household members of suspected cases were tested by RT-PCR on nasopharyngeal (NP) swabs at The Royal Children's Hospital or by nasal RAT at home. Confirmed SARS-CoV-2 cases and household members were invited to participate. Blood samples were collected approximately one month following PCR/RAT diagnosis.

### SARS-CoV-2 ELISA method

A SARS-CoV-2 S1 in-house ELISA was used to measure IgG antibody responses. Details of the ELISA method has been previously published[11,19]. Briefly, 96-well high-binding plates were coated with S1 (Sino Biological, China) antigen diluted in PBS at $2 \mu g/mL$. Goat anti-human IgG-horseradish peroxidase (HRP) conjugated secondary antibody (1:10,000; Southern Biotech; USA) was used, and the plates were developed using 3.3′, 5.5′-tetramethylbenzidine substrate solution. The ELISA method was previously validated against two commercial assays (Diasorin LIAISON® and Wantai) and a SARS-CoV-2 micro neutralization assay[11,19]. The variant-specific S1 antigens (Delta and Omicron) were sourced from Sino Biological (Sino Biological Inc., China https://www.sinobiological.com). Seropositive samples were titrated and calculated based on the World Health Organization SARS-CoV-2 pooled serum standard (NIBSC code: 20/130, National Institute of Biological Standards and Controls, UK). Results are reported in binding antibody units per millilitre (BAU/mL). The cutoff for seropositivity was 8.82 BAU/mL based on pre-pandemic samples, whereas seronegative samples were given half of the seropositive cutoff value.

### SARS-CoV-2 surrogate viral neutralization test (sVNT)

SARS-CoV-2 surrogate virus neutralization test (sVNT) was conducted to measure the neutralizing antibody responses according to the manufacturer's instructions (GenScript, New Jersey, USA). All serum samples were measured at a 1:10 dilution. For Omicron sVNT, an

Omicron specific HRP-conjugated RBD was used instead of the Wuhan HRP-RBD, and the Omicron-specific neutralizing antibody standard was used as the positive control. The results were reported as a percentage (%) of inhibition calculated from the formula; % inhibition = [1-(optical density value of sample/optical density value of the background)] × 100%. Results <30% inhibition are considered negative, while ≥30% inhibition are considered positive for neutralizing antibodies.

## Statistical analysis

Proportion of participants who seroconverted between different variants were compared using Fisher's Exact test. Antibody concentrations and sVNT neutralizing antibodies were reported as geometric mean concentration (GMC) and percent inhibition, respectively. The S1-specific IgG antibody concentrations (BAU/ml) and neutralizing antibodies (%) between children from different SARS-CoV-2 waves, between unvaccinated and vaccinated, as well as between different age groups were compared using Mann–Whitney U test. For comparison of variant-specific antibody responses, Wilcoxon-signed rank test were used. All analyses were performed with GraphPad Prism version 9.0 (GraphPad Software, https://www.graphpad.com). A $p < 0.05$ was considered significant.

## Reporting summary

Further information on research design is available in the Nature Portfolio Reporting Summary linked to this article.

## Data availability

All data generated or analyzed during this study are included in this published article (and its supplementary information files). Source data are provided with this manuscript. Source data are provided with this paper.

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

## Acknowledgements

We thank the study participants and families for their involvement in this study. We also acknowledge the SAEFVIC Research Team (Alissa McMinn, Hayley Giuliano, Kate Hession, Belle Overmars, Chelsea Bartel). We also acknowledge the Murdoch Children's Research Institute (MCRI) Biobanking service for their help in processing the samples. Funding for the recruitment of participants was provided by the Royal Children's Hospital Foundation and the MCRI Infection and Immunity Theme, grants from the DHB Foundation and the Victorian Department of Jobs, Precincts, and Regions. This work is supported by Victorian Government's Medical Research Operational Infrastructure Support Programme. MCRI. P.V.L. is supported by NHMRC Career Development Fellowship. D.B., A.C.S., and N.C. are supported by NHMRC Investigator Grants. A.C.S. is supported by a Viertel Senior Medical Research Fellowship. S.T. is supported by a MCRI Clinician Scientist Fellowship. Z.Q.T. is supported by an MCRI Early Career Fellowship.

## Author contributions

Concept and design: Z.Q.T., N.M., J.N., N.W.C., S.T., and P.V.L. Acquisition, analysis, or interpretation of data: Z.Q.T., N.M., J.N., J.A., L.A.H.D., R.A.H., N.W.C., S.T., and P.V.L. Drafting of the manuscript: Z.Q.T., N.M., J.A., R.A.H., J.N., S.T., and P.V.L. Critical revision of the manuscript for important intellectual content: Z.Q.T., N.M., J.A., D.P.B., N.C., A.C.S., N.W.C., K.M., S.T., and P.V.L. All authors approved the final version of this manuscript.

## Competing interests

N.W.C. received funding from the National Institute of Health for influenza and COVID-19 research. All other authors reported no competing interests.
