## [Peer Review File · Nature Communications]

Comparison of antibody responses to SARS-CoV-2 variants in Australian childrenReviewers' comments:

Reviewer #1 (Remarks to the Author):

This study report describes serologic studies of pediatric serum specimens collected as part of an ongoing household cohort study. They present data on the antibody response after infection in unvaccinated children infected with Wuhan, Delta or Omicron variants and vaccinated children with breakthrough Omicron infection. They assess binding IgG antibodies with a SARS-CoV-2 S1-specific IgG assay and neutralizing antibodies with a surrogate virus neutralization test (sVNT). They conclude that Omicron infection in children results in lower antibody levels and function compared with Wuhan or Delta infection or with vaccinated children with breakthrough Omicron infection. The description of the data needs to be more complete to understand the data presented.

1. The investigators note using variant-specific IgG assays and Wuhan or Omicron neutralization assays. They need to state which assays were used for which data in tables, figures, and text. They should provide more detail on the surrogate neutralization assay, is it rbd binding inhibition, pseudovirus neutralization assay, or other.
2. Figure 1 presents BAU, binding antibody units. Though in the supplementary material it would be useful to briefly note how these units are determined.
3. On p. 4, lines 77-79, the authors state, "Among the 56 children infected with the Wuhan strain, 21 (37.5%) seroconverted by day 36, while all 35 children infected with Delta ($P < 0.0001$), and 13 of 16 (81%) infected with Omicron ($P = 0.004$) variants seroconverted (Figure 1A)". This is a lower seroconversion rate for Wuhan strain than others have reported. They should state in the text which assay was used for these results. Do the authors have thoughts on why? For example, which antigen was used in the assay and is it possible there was a problem with the antigen? These data also seem inconsistent with their overall conclusion that the response to Omicron infection is less than Wuhan or Delta infection.
4. In the legend for figure 1, the authors state, "(A) SARS-CoV-2 Immunoglobulin G (IgG) Seropositivity Rate at Day 36 post-infection. Among children who seroconverted at Day 36 post-infection". Panel A, Figure 1 gives percent positivity and not data for just those that "seroconverted at Day 36 post-infection."
5. I did not find legends for the "extended data figures" in the supplement. Unless I missed, legends are should be added.

Reviewer #2 (Remarks to the Author):

Major comments

- 1) This does not appear to be a cohort study. It looks to be a combination of two studies, one that is a case-ascertained study (not a household cohort study) and the other appears to be more of a case series of children infected with omicron that had been vaccinated. Please clarify the methods. This influenza paper might be helpful to assess whether this is a case-ascertained study in part (PMCID: PMC3883904). Case ascertained studies differ from cohorts in that you are partially enrolling participants based on their outcome (they already have influenza) and partially on exposure (they are a household member being exposed to influenza) and are a design that is optimized to study transmission.
- 2) The usefulness of this study is limited by the fact that there are no pre-infection samples. Children in later periods are more likely to have been previously infected.
- 3) Differences in antibody response are likely related in part to the strength and length of the exposure. That is that individuals that were infected for longer (often measured by duration of viral shedding) or were sicker, are likely to have a stronger immune response. Can the authors provide information on these factors and perhaps adjust for them?
- 4) I am struggling with the data that so few children did not seroconvert following Wuhan infection (37.5%). Is it possible that some of these were false positives? This number does not match the literature.
- 5) I know it is in a way standard practice to separate out seropositivity from antibody levels in those

that were seropositive, however, at a population level it is really the population level antibodies that matter. That is, the average level of antibodies taking into account those that have none.

Minor Comments

- 1) Providing the range of collection times in addition to the median time for the samples would be very helpful to evaluate if these collections are similar.
- 2) Methods state that both PCR and RAT were used. However, in the extended table it states time since PCR test. Please clarify.
- 3) Authors use seroconversion or serconvert throughout, but since there are no pre samples, they are really looking at seropositivity and should use that term.

We thank the Reviewers and Editor for their careful review of our manuscript. A number of points raised by the Reviewers related to information that was contained in the Extended Data section given the limitations of the Brief Communication format. We have now moved most of this information into the main manuscript and formatted it accordingly.

We have responded to the queries raised by each Reviewer which have strengthened our manuscript. These changes are documented as tracked changes in the revised version of the manuscript.

Reviewer #1:

This study report describes serologic studies of pediatric serum specimens collected as part of an ongoing household cohort study. They present data on the antibody response after infection in unvaccinated children infected with Wuhan, Delta or Omicron variants and vaccinated children with breakthrough Omicron infection. They assess binding IgG antibodies with a SARS-CoV-2 S1-specific IgG assay and neutralizing antibodies with a surrogate virus neutralization test (sVNT). They conclude that Omicron infection in children results in lower antibody levels and function compared with Wuhan or Delta infection or with vaccinated children with breakthrough Omicron infection. The description of the data needs to be more complete to understand the data presented.

1. *The investigators note using variant-specific IgG assays and Wuhan or Omicron neutralization assays. They need to state which assays were used for which data in tables, figures, and text. They should provide more detail on the surrogate neutralization assay, is it rbd binding inhibition, pseudovirus neutralization assay, or other.*

This information was included in the Supplementary methods section and was indicated in the relevant Figures and/or Figure legends. We have now added the Methods section to the main manuscript and included this information on Pages 7-9, lines 158-208 of the revised manuscript.

2. *Figure 1 presents BAU, binding antibody units. Though in the supplementary material it would be useful to briefly note how these units are determined.*

This information was included in the Supplementary methods section. However, we have now added this information in the main manuscript under the Methods section on Page 8, lines 183-186 of the revised manuscript.

3. *On p. 4, lines 77-79, the authors state, “Among the 56 children infected with the Wuhan strain, 21 (37.5%) seroconverted by day 36, while all 35 children infected with Delta ($P < 0.0001$), and 13 of 16 (81%) infected with Omicron ($P = 0.004$) variants seroconverted (Figure 1A)”. This is a lower seroconversion rate for Wuhan strain than others have reported. They should state in the text which assay was used for these results. Do the authors have thoughts on why? For example, which antigen was used in the assay and is it possible there was a problem with the antigen? These data also seem inconsistent with their overall conclusion that the response to Omicron infection is less than Wuhan or Delta infection.*

Figure 1A was used based on Wuhan antigen to compare across the different cohorts. We have included this information on Page 4, line 82 and in the Figure legends, Page 11, lines 282-284 of the revised manuscript. The other data in Figures 1B-D include information related to which antigen was used for these assays.

The lower seroconversion for Wuhan is most likely due to the lower viral loads associated with this strain compared with Delta and Omicron variants, although we did not measure this directly as the swabs were not available. The lower seroconversion rate for Wuhan strain in children is likely due to a combination of more robust innate and/or mucosal immunity, which was highlighted in our previous paper (Toh et al., JAMA Netw Open, 2022). In this paper, the lower response to Omicron is based on the antibody concentrations (BAU/ml) and surrogate virus neutralization (sVNT) measured in Figures 1B-D, therefore it is unlikely that there was a problem with the antigen. Our conclusion was based on those who seroconverted, which shows lower antibody concentrations following Omicron infection when compared with antibody concentrations in children following Wuhan or Delta infection.

4. *In the legend for figure 1, the authors state, “(A) SARS-CoV-2 Immunoglobulin G (IgG) Seropositivity Rate at Day 36 post-infection. Among children who seroconverted at Day 36 post-infection”. Panel A, Figure 1 gives percent positivity and not data for just those that “seroconverted at Day 36 post-infection.”*

We agree that this sentence in the legend needs to be revised, apologies for the confusion. Panels B-D relate to data only for those children who seroconverted. We have revised the figure legend to: “SARS-CoV-2 Immunoglobulin G (IgG) Seropositivity Rate at Day 36 post-infection as measured by IgG ELISA using the Wuhan S1 antigen (A). SARS-CoV-2 IgG concentrations as measured by ELISA using the Wuhan S1 antigen (B), SARS-CoV-2 neutralising antibodies as measured by surrogate virus neutralisation test using the Wuhan-specific antigen (C), SARS-CoV-2 neutralising antibodies as measured by surrogate virus neutralisation test using Wuhan- or Omicron-specific antigen (D), for those children who seroconverted at Day 36 post-infection.”

5. *I did not find legends for the “extended data figures” in the supplement. Unless I missed, legends are should be added.*

All figure legends, including the extended data figures, were included at the end of the manuscript, Pages 11-12 of the revised manuscript.

Reviewer #2:

Major comments

- 1) *This does not appear to be a cohort study. It looks to be a combination of two studies, one that is a case-ascertained study (not a household cohort study) and the other appears to be more of a case series of children infected with omicron that had been vaccinated. Please clarify the methods. This influenza paper might be helpful to assess whether this is a case-ascertained study in part (PMCID: PMC3883904). Case ascertained studies differ from cohorts in that you are partially enrolling participants based on their outcome (they already have influenza) and partially on exposure (they are a household member being exposed to influenza) and are a design that is optimized to study transmission.*

We agree with the reviewer that this is a case-ascertained study and have revised this in the Methods on Pages 7-8, lines 167-175 of the revised manuscript. The main message of our paper was that children with Omicron infection generated lower antibody responses than children infected with Wuhan or Delta, and that this lower response can be circumvented by prior vaccination. Our secondary message was that measuring antibody response to Wuhan antigen in children infected with Omicron (as is done with many major commercial assays at present) is likely to substantially underestimate the antibody responses as compared with measuring using Omicron antigen.

- 2) *The usefulness of this study is limited by the fact that there are no pre-infection samples. Children in later periods are more likely to have been previously infected.*

This is a valid point. In Australia, and in particular the state of Victoria where this study was undertaken, severe and prolonged lockdown policies were implemented throughout most of 2020 and 2021 leading to much lower exposure rates among children and adults compared to other countries such as the UK and USA. Therefore, the seroprevalence among children prior to Omicron infection is likely to be very low. This was recently shown in the publication by Koirala et al. (Med J Aust, 2022), reporting a seroprevalence of 0.42% among Australian children. Conversely, if a substantial proportion of children in our Omicron cohort had a prior SARS-CoV-2 infection, this would have generated much higher levels of antibodies than we actually observed. However, we have added this as a limitation on Pages 6-7, lines 145-147 of the revised manuscript: “We did not have pre-infection blood samples to rule out prior infection but we believe this to be unlikely given the low SARS-CoV-2 exposure reported among Australian children prior to Omicron infection”.

- 3) *Differences in antibody response are likely related in part to the strength and length of the exposure. That is that individuals that were infected for longer (often measured by duration of viral shedding) or were sicker, are likely to have a stronger immune response. Can the authors provide information on these factors and perhaps adjust for them?*

We agree with Reviewer 2 about this point and included this in the Discussion (Page 6, lines 125-129 of the revised manuscript). However, we did not measure viral loads in these children as mentioned above and included this as a limitation. Our previous paper (Toh et al, JAMA Network Open 2022) showed that antibody levels were not affected by duration of viral shedding. Moreover, our cohort was a non-hospitalized cohort with mostly mild or asymptomatic infection so we do not believe that this was a major contributing factor to this result.

- 4) *I am struggling with the data that so few children did not seroconvert following Wuhan infection (37.5%). Is it possible that some of these were false positives? This number does not match the literature.*

These data were published in Toh et al JAMA Network Open (2022) comparing responses between children and adults infected with Wuhan only. This paper did a detailed analysis of factors that may have been associated with this response, including viral load, viral shedding, age and clinical symptoms but found no associations. We believe that during the Wuhan wave, a majority of children were protected through a combination of enhanced innate immune and local (mucosal) immune responses which cleared the virus more efficiently, preventing the establishment of infection and subsequent seroconversion. A recent paper also documented lower antibody responses in children compared with adults with mild or asymptomatic COVID-19 in line with these findings (Hachim et al. Nat Comms, 2022). We did not observe this for Delta or Omicron likely due to the higher viral loads reported for these variants. We have included this information in the Discussion on Page 6, lines 117-126 of the revised manuscript.

- 5) *I know it is in a way standard practice to separate out seropositivity from antibody levels in those that were seropositive, however, at a population level it is really the population level antibodies that matter. That is, the average level of antibodies taking into account those that have none.*

We agree that population level immunity is important, which is why we think our findings of low antibody levels induced by Omicron infection in children is novel and important. It is not known how well these low antibody levels protect against reinfection and their implications for children's long-term health and wellbeing.

Thankyou for this important point. Our paper is focused on comparison of antibody concentrations and function in children who were seropositive across the different SARS-CoV-2 waves. The population level of seropositivity is a recognized approach to track differences in exposure over time for seroprevalence studies, indicated in Fig 1A. We have not added a comparison of antibody concentrations across all children (including those who were seronegative) as we do not think this is necessary for this analysis, however we can add this as Extended Data if the journal request this.

Minor Comments

- 1) *Providing the range of collection times in addition to the median time for the samples would be very helpful to evaluate if these collections are similar.*

This information was included in the Extended Data Table 1.

- 2) *Methods state that both PCR and RAT were used. However, in the extended table it states time since PCR test. Please clarify.*

Thankyou for pointing this out, we have corrected this in the Extended Data Table 1.

- 3) *Authors use seroconversion or serconvert throughout, but since there are no pre samples, they are really looking at seropositivity and should use that term.*

We have now used seropositivity as suggested.

REVIEWERS' COMMENTS

Reviewer #1 (Remarks to the Author):

The authors have responded to comments. A few additional comments.

1. It would help the reader understand the data if more detail was included in the figure labels, e.g. Figure 1 it would help to include the S variant used for figure C and D.
2. In the methods, the dilution(s) tested for the neutralization assay should be stated and more detail on how % inhibition is determined should be included.
3. The term extended data was confusing to me until I realized it indicates, supplemental data.

Reviewer #2 (Remarks to the Author):

The authors have sufficiently addressed my concerns.

Reviewer #1:

The authors have responded to comments. A few additional comments.

- 1. It would help the reader understand the data if more detail was included in the figure labels, e.g. Figure 1 it would help to include the S variant used for figure C and D.*

Thank-you for this point. This information was included in the Figure legend, but we have now made this clearer. Data in Figure 1C was based on Wuhan RBD antigen while for Figure 1D, the relevant RBD antigen used is indicated in the Figure labels.

- 2. In the methods, the dilution(s) tested for the neutralization assay should be stated and more detail on how % inhibition is determined should be included.*

We have now added further details on the neutralization assay in the methods (refer Page 9, lines 201-206 of the revised manuscript).

- 3. The term extended data was confusing to me until I realized it indicates, supplemental data.*

Thank-you for this point. We have now changed this to Supplementary Data.